# Rapid emergence of climate change in environmental drivers of marine ecosystems

Stephanie A. Henson[1], Claudie Beaulieu[2], Tatiana Ilyina[3], Jasmin G. John[4], Matthew Long[5], Roland Séférian[6], Jerry Tjiputra[7] & Jorge L. Sarmiento[8]

Climate change is expected to modify ecological responses in the ocean, with the potential for important effects on the ecosystem services provided to humankind. Here we address the question of how rapidly multiple drivers of marine ecosystem change develop in the future ocean. By analysing an ensemble of models we find that, within the next 15 years, the climate change-driven trends in multiple ecosystem drivers emerge from the background of natural variability in 55% of the ocean and propagate rapidly to encompass 86% of the ocean by 2050 under a 'business-as-usual' scenario. However, we also demonstrate that the exposure of marine ecosystems to climate change-induced stress can be drastically reduced via climate mitigation measures; with mitigation, the proportion of ocean susceptible to multiple drivers within the next 15 years is reduced to 34%. Mitigation slows the pace at which multiple drivers emerge, allowing an additional 20 years for adaptation in marine ecological and socio-economic systems alike.

[1] National Oceanography Centre, European Way, Southampton, SO14 3ZH, UK. [2] Ocean and Earth Sciences, University of Southampton, European Way, Southampton, SO14 3ZH, UK. [3] Max Planck Institute for Meteorology, Bundesstr. 53, D-20146 Hamburg, Germany. [4] NOAA/Geophysical Fluid Dynamics Laboratory, 201 Forrestal Road, Princeton, New Jersey 08540, USA. [5] Climate and Global Dynamics, National Center for Atmospheric Research, PO Box 3000, Boulder, Colorado 80307, USA. [6] Centre National de Recherches Météorologiques, Météo-France/CNRS, 42 Avenue Gaspard Coriolis, 31057 Toulouse, France. [7] Uni Research Climate, Bjerknes Centre for Climate Research, Box 7803, NO-5020, Bergen, Norway. [8] Atmospheric and Oceanic Sciences Program, Princeton University, 300 Forrestal Road, Sayre Hall, Princeton, New Jersey 08544, USA. Correspondence and requests for materials should be addressed to S.A.H. (email: S.Henson@noc.ac.uk).

Marine ecosystems provide services of high socio-economic value[1], including the primary protein source for one in seven of the world's population[2] and regulation of Earth's climate via the uptake and storage of atmospheric carbon dioxide[3,4]. However, climate change is predicted to have profound consequences for marine ecosystems, affecting both their structure and functioning, for example, refs 5–7. The Intergovernmental Panel on Climate Change (IPCC) identifies four principal climate drivers that affect marine ecosystem structure, functioning and adaptive capacity[8]: pH, temperature, oxygen concentration and food availability. All four are subject to substantial perturbations in projections of future climate change scenarios.

Research has generally focused on the potential negative effects of climate change on marine ecosystems. For example, ocean pH is reduced by increasing atmospheric $CO_2$ concentration, which may result in reduced viability of calcareous organisms, among other effects[5]. Warming ocean temperatures are associated with increased ocean stratification, which restricts nutrient supply to photosynthetic organisms in surface waters[9]. The solubility of oxygen and exchange of subsurface waters with the atmosphere will also be reduced with warmer temperatures, driving lower oceanic oxygen concentrations with potentially negative effects on marine organisms[10]. Although regionally variable, the combined effect of these changes is predicted to be an overall global decrease in primary production (PP), which is the ultimate determinant of food availability to marine ecosystems[11]. However, in some cases positive (or neutral) responses to potential marine stressors have been observed[12,13], implying that uncertainty surrounding the future of the marine ecosystem is large. Here we adopt the terminology that a change in a potential stressor where the ecosystem response is unknown (and may not necessarily be negative) is termed a 'driver'.

Although the projected climate change response over the coming century in these environmental drivers is large, so is the natural variability encountered by marine organisms, suggesting that some species have the capacity to adapt or acclimate to change[5,14,15]. Importantly though, natural variability occurs on timescales of a few years (interannual variability) to millennia (glacial–interglacial cycles), whereas climate change is essentially a one-way street, so that associated changes in the marine environment are unlikely to be reversed. Hence, climate change will eventually push marine ecosystem drivers beyond the range of natural variability, potentially resulting in migration of species[16], reorganization of ecological niches, the establishment of novel climates[14,17] and the requirement for socio-economic systems to adjust to these changes so that livelihoods and human well-being are protected.

In addition, marine ecosystem drivers rarely vary in isolation and multiple factors may act additively or synergistically to increase the impact of a single driver[18–20]. For example, ocean acidification may alter the carbon to nitrogen ratio of sinking organic material so that more oxygen is required for remineralisation[21,22]. Ocean acidification may act in concert with rising temperatures to reduce coccolithophore abundance or calcite production[23]. In turn, lower pH and oxygen concentration can enhance temperature sensitivity in corals[24,25] and crustaceans[26]. The synergistic effects of multiple drivers are challenging to investigate in field or lab studies, due to the difficulties of undertaking multi-factorial experiments over multiple generations. However, existing observational evidence tends to suggest that co-occurring stress can add to or amplify the effect of a single stressor[18], as seen in a microalgal species grown in 96 experiments where population growth declined with the number of environmental drivers[27]. Other studies have shown, however, that multiple stressors can interact in unexpected ways, sometimes resulting in a positive or neutral effect[12,28]. For example, warming and ocean acidification have antagonistic effects on sea urchin larval growth, resulting in minimal overall impact (although both have a negative effect on larval abnormality[29]).

Although the response of the marine ecosystem to changing drivers is not yet clear, it is nevertheless important to quantify when, where and which combinations of stressors are likely to occur. Of particular relevance for an organism's ability to adapt to a changing climate is the speed with which drivers of ecosystem stress emerge from the background of natural variability[30–32]. The environmental niche that organisms occupy roughly matches the ambient conditions that they experience[33,34], that is, the organisms must be resilient to the range in natural variability. Organisms must be adapted to at least the annual extrema in conditions during winter and summer (the seasonal variability), which in almost all locations exceeds interannual variability. The more rapidly the system is pushed out of its natural range of variability, the less time the organisms will have to adapt or acclimate to the new conditions or migrate to more suitable areas. Where rapid emergence of multiple drivers occurs, marine ecosystems and dependent socio-economic systems may be unable to adjust sufficiently quickly to avert disruption.

Here we examine how rapidly climate change signals in the annual extrema of the drivers of ecosystem stress emerge from the background of natural variability in several ocean properties (sea surface temperature (SST), pH, PP and interior oxygen content), focusing on when and where multiple drivers occur simultaneously. Further, we investigate whether mitigation efforts are able to slow the pace at which multiple drivers emerge. We quantify when the climate change signal exceeds natural variability in the principal drivers of marine ecosystem stress identified by the IPCC under two climate change scenarios[35]: a 'business-as-usual' scenario (RCP8.5)[36,37] and a mitigation scenario (RCP4.5). The mitigation scenario was chosen as representative of a possible post-COP21 situation, as a result of which nations submitted Intended Nationally Determined Contributions to indicate their intended level of emissions post 2020. Even if achieved, these intended cumulative emissions still imply a median warming of 2.6–3.1 °C by 2100 (ref. 38), which is within the range of RCP4.5. Currently, global emissions are tracking along the upper end of the IPCC scenarios, as represented by RCP8.5 (refs 39,40).

## Results

**Time of emergence.** Using a multi-model ensemble from the CMIP5 archive (Supplementary Table 1), we construct time series for each variable of annual maxima (for SST) or minima (for pH, PP and oxygen) for the period 1860–2100 using a combination of the historical runs (1860–2005) and future scenarios (2006–2100)[41]. The start of the climate change signal is defined as the year when conditions become persistently uni-directional, and the time of emergence (ToE) is defined as the year when annual extrema exceed the long-term trend persistently for the remainder of the time series. The long-term trend is presented in Supplementary Note 1; the 'Methods' section and Supplementary Fig. 1 provide more details on ToE definition. The ToE presented here thus represents the exposure of the ecosystem to conditions outside the range of previously experienced seasonal variability. In contrast to previous work on emergence of marine stressors, for example, refs 11,42–44, we define natural variability using the seasonal amplitude derived from monthly model output. (The same analysis performed on annual mean model output results in earlier ToE and more rapid pace of climate change; Supplementary Note 2 and Supplementary Figs 2–6). We recognize that variability on shorter timescales, such as diurnal

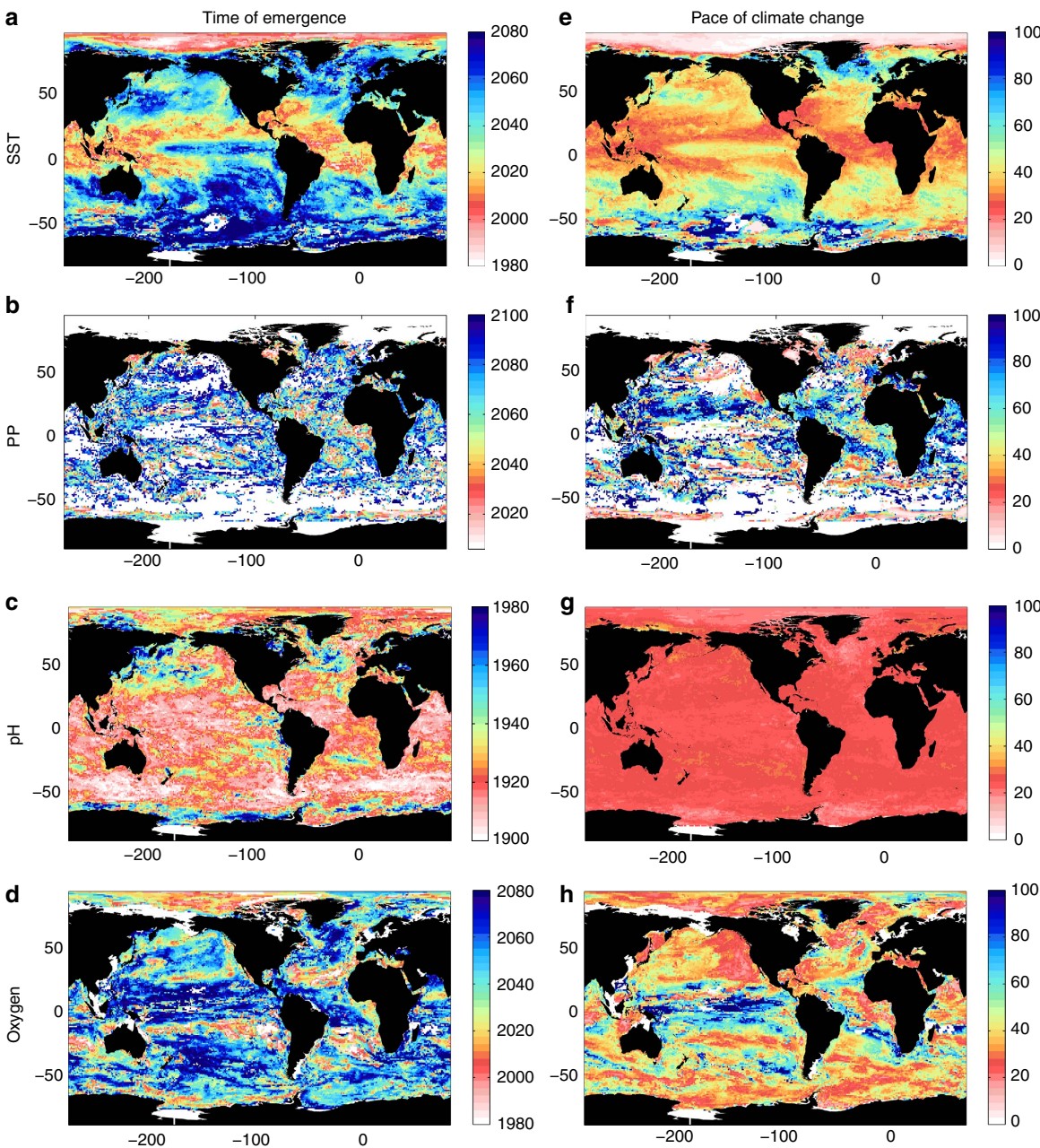

**Figure 1 | ToE and pace of climate change in ecosystem drivers.** Multi-model median of the year when annual extrema exceed the climate change trend (see 'Methods' section) for (**a**) SST, (**b**) PP, (**c**) pH and (**d**) interior oxygen content in the 'business-as-usual' scenario (RCP8.5). Note the different colour scales for each variable. (**e**–**h**) The pace of climate change: the number of years between the start of climate change and the signal emerging (see 'Methods' section). White areas indicate where ecosystem stress does not emerge above the range of variability for that parameter by 2100.

or extreme event-related (for example, heatwaves, storms), can also be pronounced and important[45], but we are constrained here by the temporal resolution of the available model output. In addition, the coarse model grid (1° × 1°) eliminates small-scale spatial variability. If natural variability is higher than we estimate (but the trend is of similar magnitude) then ToE will be later than we calculate.

The ToE is illustrated in Fig. 1a–d. The climate change signals of pH and SST emerge very rapidly (global median of 1924 (± 4.7 years) for pH and 2034 (± 8.8 years) for SST; ± = inter-model s.d. on the global median). Indeed, the climate change signal in pH already exceeds the bounds of natural variability (ToE < 2016) in 99% (± 0.5%) of the open ocean. The climate change signal in SST has also already emerged in the

subtropics and the Arctic. The climate change trends in PP and interior oxygen content emerge later (global median of 2052 (± 12 years) for oxygen and 2070 (± 8.3 years) for PP).

The pace of climate change is represented here by the length of time (in years) between the start of the climate change signal and its emergence from the background natural variability (Fig. 1e–h). The pace of change is uniformly very rapid in pH, occurring in ∼ 25 years almost everywhere (see also Supplementary Fig. 7). pH undergoes both a rapid pace of change and emerges early, due to its very small interannual variability, and because a large fraction of anthropogenic $CO_2$ has been absorbed by the ocean[46]. (Note however that in coastal waters, which are not resolved by these global models, variability in pH is substantially greater[47]). For SST, the pace is most rapid (< 25 years) in subtropical

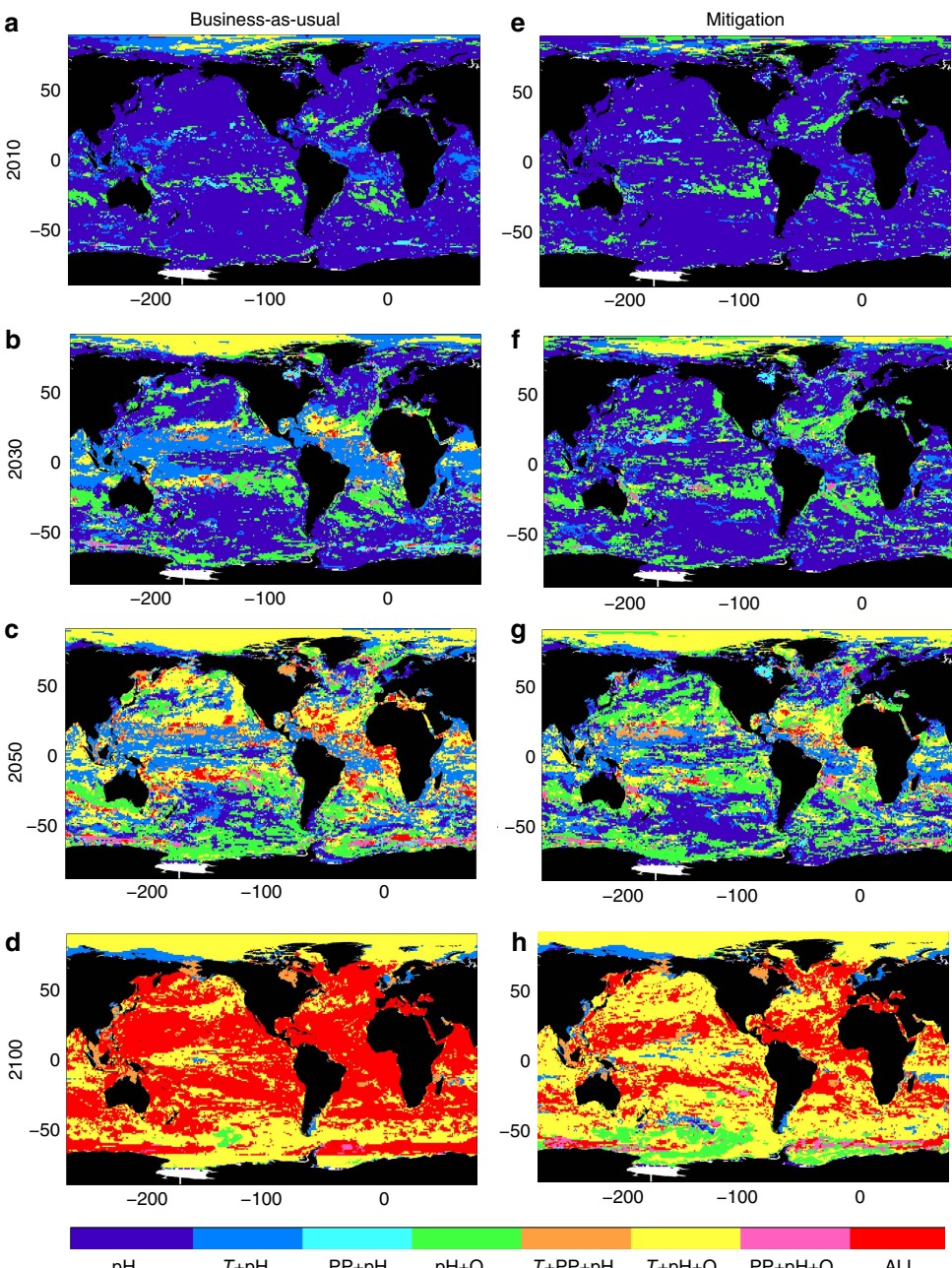

**Figure 2 | Emergence of multiple drivers.** Combination of stressors that have emerged above the background of seasonal variability by (**a**) 2010, (**b**) 2030, (**c**) 2050 and (**d**) 2100 for a 'business-as-usual' scenario (RCP8.5), based on the model mean ToE estimates shown in Fig. 1. (**e–h**) same, but for the mitigation scenario (RCP4.5). In the legend, $T$ refers to SST, PP to primary production and $O_2$ to interior oxygen concentration.

and Arctic regions, and considerably slower in subpolar areas (>50 years). Interior oxygen content evinces a slower pace of change, with the exception of the Northeast Pacific, where models likely underestimate the large natural variability associated with major climate modes[48]. PP shows the slowest pace of change, potentially because PP acts as an integrator of changes in light, temperature and nutrients. The later ToE in PP is consistent with previous observations that trends are more rapidly detectable in parameters such as SST and pH than in PP, primarily due to the large natural variability in the latter[49,50].

**Emergence of multiple drivers.** Where the rapid emergence of multiple drivers co-occurs is where ecosystems are likely to experience the greatest exposure to climate change[20,51]. Here we

demonstrate that by 2030 regions encountering multiple drivers, principally by pH and SST, dominate the ocean (Fig. 2a–d). Species sensitive to change in pH and/or temperature will likely need to adapt to these new conditions very rapidly. In particular, regions of low variability (for example, subtropics) will potentially be more vulnerable to climate change as these areas emerge both earliest and most rapidly. However, although ocean acidification and warming are widespread, the climate change signal in PP and oxygen remains smaller than the substantial natural variability in most regions in the near future. The exception is the Arctic, which appears to be a hotspot of change with rapid emergence of the climate change response in pH, SST and oxygen content. (However, note that the Arctic Ocean is also a region of high model uncertainty in projections of PP[44,52]).

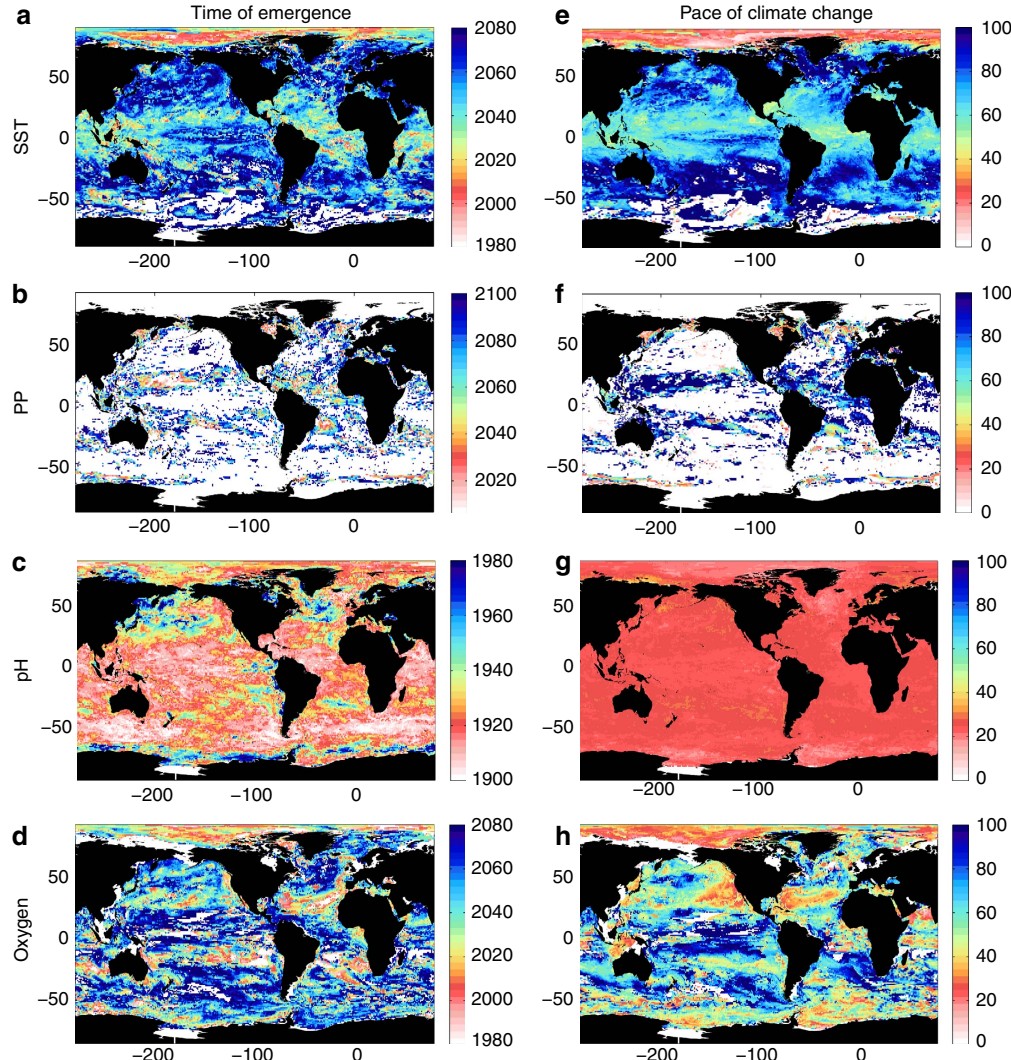

**Figure 3 | ToE and pace of climate change in ecosystem drivers under a mitigation scenario.** Multi-model median of the year when climate change trend exceeds the range of natural seasonal variability (see 'Methods' section) for (**a**) SST, (**b**) PP, (**c**) pH and (**d**) interior oxygen content in a mitigation scenario (RCP4.5). Note the different colour scales for each variable. (**e**–**h**) Number of years between the start of climate change and the signal emerging (see 'Methods' section). White areas indicate where ecosystem stress does not emerge above the range of seasonal variability for that parameter by 2100.

By 2050, a mosaic of multiple environmental drivers has emerged (Fig. 2c). Where previously climate change trends in pH and SST dominated the patterns of environmental change, a more heterogeneous patchwork of multiple drivers now develops. In 86% (±10%) of the ocean, multiple drivers have emerged, implying the potential for widespread disruption to marine ecosystems. By 2100 an ocean environment has developed in which climate change trends have emerged in all four marine drivers of ecosystem stress in 62% (±5%) of the ocean (Fig. 2d), and a further 37% (±4%) of the ocean is experiencing a novel combination of conditions in pH, oxygen content and SST.

**The effect of mitigation.** As we have shown, following a 'business-as-usual' emissions pathway (RCP8.5) results in rapid emergence of multiple drivers of the marine environment. The aim of mitigation activities is to reduce the magnitude and rate of climate change impacts. To what extent then can mitigation limit or slow the pace of ecosystem stress emergence? Projections with the IPCC RCP4.5 mitigation scenario demonstrate that the climate change trend in marine ecosystem drivers

is reduced (Supplementary Figs 8 and 9). As a result, mitigation slows and delays the emergence of the climate change signal by several years (Fig. 3 and Supplementary Fig. 6); for example, the SST trend emerges in 2050 (±10.5 years). For PP, mitigation results in only small regions of the ocean experiencing climate change-driven conditions outside the range of natural variability before 2100 (Fig. 3).

The effect of mitigation on the emergence of multiple ecosystem drivers is clear (Fig. 4). Within the next 15 years, only 34% (±9%) of the ocean becomes susceptible to multiple drivers in a mitigation scenario[41], compared with 55% (±12%) under a 'business-as-usual' scenario (Fig. 4). Note, however, that ocean acidification is unavoidable; although reduced in a mitigation scenario, it has already emerged as a driver across the entire open ocean. The pronounced reduction in multiple stress conditions due to mitigation persists through 2050 with only 69% (±12%) of the ocean exposed, compared with 86% (±10%) under the 'business-as-usual' scenario. Even by 2100, in the mitigation scenario only 30% (±4%) of the ocean is affected by all four ecosystem drivers, in contrast to the 62% (±5%) in the 'business-as-usual' scenario, and there exist large areas where PP remains

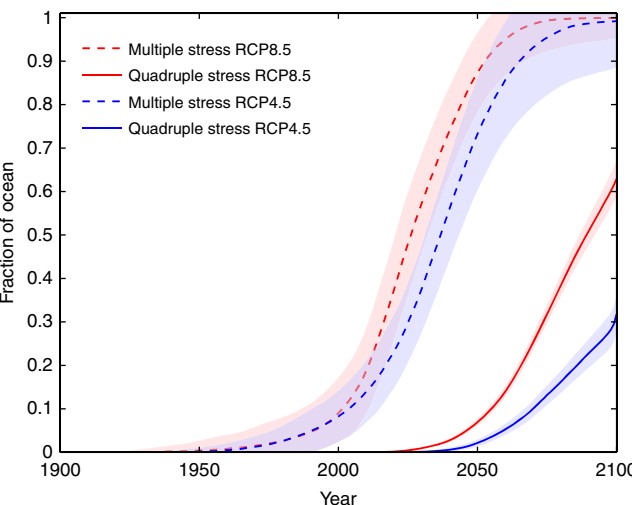

**Figure 4 | Effect of mitigation on the global emergence in drivers of ecosystem stress.** The proportion of the ocean in each year 1900–2100 affected by multiple stress (>1 driver) and quadruple stress (all 4 drivers) in the 'business-as-usual' scenario (RCP8.5) and a mitigation scenario (RCP4.5). Shaded areas represent ±1 inter-model s.d.

within the range of natural variability. Mitigation slows the pace of potential ecosystem exposure substantially, delaying the widespread emergence of the 'quadruple whammy'[20] by ~25 years (Fig. 4). The exception is the Arctic, where mitigation does little to slow the emergence of multiple drivers (Fig. 2).

## Discussion

The vulnerability of the marine ecosystem to climate change is considered to be a function of exposure and sensitivity to stressors, combined with adaptive capacity[53,54]. In this study, we quantify one part of this equation—the potential for exposure to environmental drivers. Marine ecosystems, and the individual organisms that make up those ecosystems, are adapted to the range of conditions they experience[33,34]. When the environment changes sufficiently that new conditions, or a new combination of conditions, emerge and persist, the organisms must adapt, migrate to more favourable areas, or face extinction. In this way, climate forcing can induce changes at the organism level, which result in changes to ecosystem structure, species interactions and food web dynamics[55,56]. If novel conditions emerge rapidly, species have less time to either adapt or migrate, potentially increasing the probability that disruption to the ecosystem will occur.

We find that the climate change-driven trend in pH already exceeds the range in natural seasonal variability over most of the ocean, as does SST in the subtropics and Arctic. For PP and interior oxygen content, although the trend is large, the natural variability is also large, resulting in later ToE. Species adapted to living in regions of low variability are likely to have relatively narrow environmental niches[8] and may be living close to their maximum tolerance[57]. Subtropical and tropical species are therefore likely to be more sensitive to the rapid emergence of climate change trends. Polar species are also particularly vulnerable as they cannot shift their geographical range northward in response to emerging drivers and so must either adapt to changing conditions or go extinct[58]. However, new ecological niches may open for species resilient to ocean acidification and warming waters and inured to the large natural variability that occurs in PP and oxygen.

Our model results suggest that seasonal minimum pH levels have been lower than the previously experienced natural range for >90 years already. This is consistent with analyses of (shorter) observational records, which suggest that trends in ocean acidification are likely to be anthropogenically driven[50,59]. However, evidence that this change has had a significant or lasting impact on marine organisms is scarce, except perhaps for warm-water coral communities[60]. Does this imply that the marine ecosystem is actually rather resilient to climate change? The key factor may be the speed with which climate change emerges in marine ecosystem drivers relative to the speed with which organisms can adapt. Individual species can seemingly adapt relatively rapidly (compared with the timescales of climate change) to new conditions. For example, a tropical reef fish was found to acclimate to acute exposure to warmer temperatures within two generations (damselfish have lifespans of >5 years[61]). Organisms with shorter lifespans, such as phytoplankton (~few days), adapt correspondingly more rapidly. Coccolitho-phores, for example, adapted to a large degree to more acidic conditions within 500 generations[62]. The fossil record can also provide some insight into the possible impacts of climate change on marine ecosystems (although the paleoclimate changed at slower rates than projected for anthropogenic climate change). During the Paleocene–Eocene Thermal Maximum for example, up to 50% of benthic foraminifera went extinct[63] and warm-water species expanded their ranges northward[64,65]. Although the limits of adaptation capacity are presently very poorly known, past extinctions at slower rates of climate change suggest that adaptation rates in some organisms are unlikely to be fast enough to keep pace, ultimately implying extinction.

We demonstrate that a heterogeneous mosaic of multiple environmental drivers develops in the next 50 years (Fig. 2c). This mosaic suggests that species resilient to change in one driver but negatively affected by another may be able to migrate to newly formed suitable habitats, provided the velocity of climate change (sensu[16]) does not outpace migration speed. In addition, some species may be able to alter their depth range so that they avoid decreases in thermocline oxygen whilst still remaining within their thermal niche (note that here we only assess changes in surface temperature). Again, little is known about the potential migration speed of marine organisms. Large motile species such as fishes and mammals are likely to be able to migrate rapidly to more favourable conditions (although locating these refugia may not be as simple as tracking a northward-moving isotherm). Smaller motile species, such as zooplankton, have also been observed to migrate in response to climate trends or variability, as in the North Atlantic where the distribution of warm-water copepod species has shifted northward in recent decades[66]. Planktonic species may be able to rely on rapid dispersal to maintain populations, whereas sessile species may not be able to migrate sufficiently rapidly to keep pace with future climate change. Returning to the Paleocene–Eocene Thermal Maximum as an analogy, mobile crustaceans avoided significant community changes[67], whereas sedentary sediment dwellers were heavily impacted[63].

Translating the emergence of climate drivers that we present here into an understanding of how the structure and functioning of the marine ecosystem may respond is an extremely challenging task. The interplay between biogeochemical stressors, including synergistic effects, adaptation and migration potential and speed, the bioclimatic envelope, organisms' climate sensitivity, non-linear responses to changing conditions, short-term acclimatization and extinction risk is complex in the extreme. Even the first step, of connecting the emergence of climate change in ocean conditions with potential stress in marine organisms, requires the implicit assumption that the

niche width of an organism scales with the local variability it experiences, and if conditions exceed that variability then a response (whether negative or positive) may occur. However, many species appear to thrive in environments that are less than optimal, for example, some marine fish and invertebrates have warmer or cooler temperature optima than the environment in which they are found[54]. In addition, the emergence of persistent, anomalously low oxygen (compared to previous seasonal variability) may not be of relevance to individual organisms if concentrations still remain above hypoxic levels, although oxygen has been found to limit animal life even at higher concentrations[8].

Currently, it is not clear whether ecosystem-wide adaptation or migration can outpace the speed at which multiple drivers emerge (Fig. 1). How individual species will fare, or how the ecosystem as the sum of its parts will fare, is poorly understood. What is clear however is that there are likely to be winners and losers in the future ocean[68]. However, lacking the ability to predict the future impact of drivers on marine ecosystems creates significant challenges to determining an appropriate course for sustainable management of the ecosystem services, such as fisheries, that they provide. The quantification of the ToE of multiple drivers presented here is an important first step in achieving an understanding of the response of marine ecosystems to future climate change.

Slowing the pace of climate change could give species more time to adapt to changing conditions or migrate to more suitable areas, potentially reducing extinction risk. Although most ecosystems have the capacity to adapt to changing conditions to some extent[14,69], for species endemic to the Arctic or sensitive to ocean acidification, there may be no refuge from climate change impacts, regardless of mitigation efforts. For other species, the timescales over which threshold changes in environmental stressors emerge will be key to determining the degree of disruption[17,30,69]. Our results demonstrate that mitigation measures substantially slow the pace of climate change (Fig. 4), and so likely ameliorate its impacts; it would also allow time for additional conservation planning and adaptation of fisheries and aquaculture, likely resulting in reduced species extinction risk[17,69–71]. Given that global $CO_2$ emissions are currently tracking the IPCC's business-as-usual scenario[39,40], timely implementation of the reductions pledged under COP21 is needed to slow the rapid development of ubiquitous multiple ecosystem stress.

## Methods

**Model output.** Output from 12 IPCC-class Earth system models run for the CMIP5 exercise driven by greenhouse gas, aerosol and ozone concentrations prescribed in a specific representative concentration pathway scenario of climate change were downloaded from the archive at http://cmip-pcmdi.llnl.gov/cmip5/data_portal.html. The models and institutions providing the output are listed in Supplementary Table 1. For all models the monthly output for 100-year sections taken from the end of the preindustrial control run, the historical run (1860–2005) and for the RCP8.5 and RCP4.5 scenarios for 2006–2100 were downloaded. The variables SST ('tos' in the CMIP archive), surface pH ('ph'), total annual vertically integrated PP ('intpp') and dissolved oxygen ('o2') in the thermocline (200–600 m depth range, as in refs 11,20) were used. All data were re-gridded onto a regular $1° \times 1°$ grid using linear nearest-neighbour interpolation. In the case of depth-resolved dissolved oxygen concentrations, monthly output is not a standard CMIP5 output. Data were therefore sourced from the model progenitors directly; however, output could not be obtained for HadGEM2-CC, HadGEM2-ES or IPSL-CM5B-LR models and so only nine models were used for the monthly oxygen analysis. Preindustrial control run output was detrended, if necessary, using a linear regression prior to further analysis. For each year of model output, the annual maxima (for SST) and minima (pH, thermocline oxygen and PP) were extracted. Supplementary Note 3 and Supplementary Fig. 10 contain a comparison of model output with observations.

**Regression analysis.** A generalized least-squares model with a first-order auto-regressive error term (AR(1)) was applied to time series of the annual maxima (SST) or minima (other variables) using R package 'nlme'[72]:

$$Y_t = \mu + \omega t + N_t,\qquad(1)$$

where $Y_t$ is the annual extreme in the variable, $\mu$ is a constant term (the intercept), $t$ is the linear trend function (here time in years), $\omega$ is the magnitude of the trend (the slope) and $N_t$ is the unexplained portion of the data, which is modelled as an AR(1) process. Trends for SST are reported as °C per decade; for PP, pH and oxygen as the % change per decade with respect to the mean of 1986–2005. The assumption of normality in the annual extrema was checked using the Lilliefors test. Approximately 80% of pixels pass the test for pH and SST, and ∼65% for PP and oxygen. The majority of pixels failing these tests are in the Arctic.

**Definition of ToE.** All analyses are performed at every model grid cell, separately for each model. To calculate the ToE, first the time series of annual extrema in the conjoined historical and warming scenario runs is created. Then, an inflection point is located by calculating the cumulative sum of the gradient in $Y_t$ ($\partial Y/\partial t$) and identifying the year when it exceeds zero (for a positive trend) or drops below zero (for a negative trend) for the remainder of the time series—we refer to this as the start of the climate change signal (see Supplementary Fig. 1). The trend in $Y_t$ is then calculated from that start point forward to 2100 using equation (1). The natural variability (or noise) is defined as one standard deviation in annual extrema of a 100-year section of the model's control run. As a consequence, the natural variability as defined here accounts for only the unforced natural variability, excluding the influence of changing incoming solar radiation or volcanoes. The ToE is then defined as:

$$\text{ToE} = (2 \cdot \text{noise})/\omega.\qquad(2)$$

This criterion ensures that the trend exceeds 95% of the values in the noise, assuming it is normally distributed, and therefore that the emerging signal is highly unusual. Any values of ToE exceeding 2100 are excluded from further analysis. Assumptions of normality and heteroskedasticity in the residuals from the trend regression were checked using the Lilliefors and Breusch–Pagan test, respectively. Approximately 5–30% of pixels fail one of the tests and, for most models and variables, these pixels are predominantly in the Arctic. Assuming a normal distribution for trends in annual extrema is reasonable given the low percentage (5–30%) of pixels where the assumption is not met. Where the assumptions are not met, the ToE criterion will not necessarily indicate that the trend exceeds 95% of the noise values, and may instead be lower. The ToE is calculated for each individual model and the median year for each pixel is then used as the ensemble mean. The inter-model difference in ToE is expressed as ±1 s.d. The pace of climate change is described here by the difference (in years) between the ToE and the start of the climate change signal. All analyses were performed in Matlab (Mathworks).

**Data availability.** The model output analysed here is publically available via the CMIP5 archive. The data used in the figures and the code used to generate them will be made available via request to the corresponding author.

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

## Acknowledgements

We acknowledge the World Climate Research Programme's Working Group on Coupled Modelling, which is responsible for CMIP, and we thank the climate modelling groups (listed in Supplementary Table 1 of this paper) for producing and making available their model output. This work was supported by Natural Environment Research Council National Capability funding to S.A.H. Contributions of T.I. and R.S. were supported by EU H2020 project CRESCENDO (grant 641816). Contribution of J.L.S. was supported by BP under the Princeton Carbon Mitigation Initiative. Contribution of JT was supported by Bjerknes Centre (BIGCHANGE) and RCN (no. 239965).

## Author contributions

S.A.H. conceived the study; S.A.H., C.B. and J.L.S. undertook the analyses; T.I., J.G.J., M.L., R.S. and J.T. contributed bespoke model output. All authors contributed to writing the manuscript.

## Additional information

**Competing financial interests:** The authors declare no competing financial interests.

