## [Peer Review File · Nature Communications]

Reviewers' comments:

Reviewer #1 (Remarks to the Author):

I have put most of my comments- both general and detailed into pdf 'post-it notes' in the attached pdf.

In summary, The author list is made up of unparalleled modelling expertise but has less knowledge of the responses - regionally and cumulatively - of marine life to changes (from a physiological and ultimately ecological perspective)

So in some respects it is a premature submission but I think that if carefully revised - and framed more closely to the COP21 scenarios - in particular the 1.5 C warming (and the recent commentary in NCLIM by Magnan et al. it could be a useful starting point for discussions amongst policy makers.

So acceptable after major revisions - that need to get across that although model projections of multiple drivers are well advanced, our understanding of the cumulative effects of multiple drivers lags far behind, as does teasing apart the individual and interactive effects and their mechanistic influence across a wide range of marine life.

Hence a more balanced viewpoint on the range of responses - detrimental, none, beneficial - for the organisms at each trophic level is needed - before we can even contemplate how they will translate ecologically

I sign my reviews - Philip Boyd

Reviewer #2 (Remarks to the Author):

Overall: This is an overall well written, designed, and performed study addressing an important and nontrivial question. It should be of interest to a broader audience and for a high-impact journal. There are a few points where the ms requires some edits and where a few open questions need attention.

Specific Points

Major

1. The language of detection / signal emergence is often unclear. For example, on line 35, it is unclear from what the trend "emerges"? On line 92, the language is "likely to appear".
2. It would help the reader to have an example time series (maybe in the SOM) to illustrate the ToE and other concepts introduced on page 5.
3. It is unclear / ambiguous (and in my view does not follow from the evidence presented in the ms) that mitigation could "avert" the ecosystem stress (l. 300). Please clarify.
4. Will the data and analysis code be available for reproduction once the ms is published?
5. How have the models been fitted (p. 11)?
6. The ms tests several assumptions and some are violated. How does this impact the probabilistic statements (e.g., l. 342)? Can these statements be tested? How large is the autocorrelation? In general, there is a considerable body of research on detecting trends in field with spatial and/or temporal correlations that the paper is mostly silent on.
7. What is the explanation for the bimodal distribution in Figures S6 (a,b)?

Minor:

1. The citation for IPCC and RCP on lines 110-112 seem to be missing. I assume they refer to Moss

et al (2010). In this case, is the 'business-as-usual' a citation? Re-reading the Moss et al (2010) paper suggests to me that interpreting RCP8.5 as business-as-usual can be problematic. Please clarify.

2. Please define "robustly" and "robust" (l. 284 and 287) or reword. This expression can have very different meanings depending on the discipline of the reader.
3. How is a "best course for sustainable management" defined (l. 285)?
4. How was the data regridded (l. 313)?
5. Is "extrema" the correct word choice (l. 324)?
6. Please add units to the right colour bar on fig 2.
7. Can the statement on l. 699 indeed be "verified"?
8. Whose / which confidence is discussed on l. 703?

Reviewer #3 (Remarks to the Author):

Review of "Rapid emergence of climate change in environmental drivers of marine ecosystem stress" by Henson et al.

General comments:

The authors use CMIP5 model output to investigate the emergence of climate change signals and the timing at which annual extremes in key variables (pH, SST, oxygen, primary production) surpasses their natural variability. They find that pH has already been changed beyond its natural variability, and so has SST in the subtropics and the Arctic. Under a business-as-usual scenario (RCP8.5), they find that most of the ocean will be subjected to multiple stressors by 2050 (the subtropics and the Arctic by 2030). However, under a mitigation scenario (RCP4.5), the emergence of multiple stressors could be delayed in large parts of the ocean by ~20 years, allowing for longer adaptation periods for ecosystems.

The study is novel and of great interest. It aims at tackling one of the issues related to the vulnerability of marine ecosystems under climate change -the exposure to multiple stressors- and also discusses the relevance of the results in terms of the time that will be available for marine ecosystems to adapt. I found the methods sound and appropriately explained and I believe the conclusions are robust; furthermore, the authors do a good job at presenting some caveats and limitations of their analysis (eg, coastal regions not included in their analysis).

As a potential addition to the manuscript, I would suggest a discussion on how current emissions compare between RCP8.5 and RCP4.5 - given that both scenarios start in 2006, there could be a description of how much closer we are to the business-as-usual scenario (maybe a good place to emphasize the risks of not reducing our emissions ASAP). Furthermore, I found quite interesting that PP seems to be much more resilient to climate change than other variables that could affect it (eg, SST, pH). Therefore, I would love to see a brief explanation about the expected variability in PP - is it that climate-induced changes in other variables act in different directions, such that PP is not so affected (e.g. warming that would enhance PP overlapped with increased stratification that reduces nutrients and would decrease PP) and/or missing processes in the models (e.g. no phytoplankton groups that actually depend on pH)? Lastly, the authors mentioned some caveats of the analysis (constrained to use monthly output and miss the potentially larger high-freq variability; the fact that global models do not resolve the coastal ocean, which are regions of higher natural variability). Could the authors discuss any other limitations due to the coarse grids of global models? (eg. missing variability due to eddies and other unresolved mesoscale processes, etc).

Overall, I found the manuscript well written and worth of publication in Nature Communications.

Specific comment:

148: "PP and interior oxygen content evince a much slower pace of change". When I look at figure 1, oxygen seems to have a more rapid pace (20-40 years) than PP, except in the tropics. Actually, figure 1h (oxygen) seems as "red" as figure 1e (SST). It does not seem appropriate to equate PP with oxygen (also done in the discussion) - even the ToE for oxygen in the Arctic and OMZ regions seems to be quite close to present date. Also, it may be worth noting that in the OMZ regions, pH+O2 emerge as stressors by 2030. I recommend adjusting the description of oxygen changes (or consider improving the description and why the authors think is comparable to PP changes).

Minor comments:

51: identify  identified? identifies?

116: could use "SST" already

183: suggest including reference to fig S8

188-189: suggest to rewrite: "undavoidable," "unavoidable;" and ", and" ", it"

196: I feel "quadruple whammy" is not a friendly phrase for non-native English speakers. I suggest re-phrasing.

198: "Fig 1" should it read Fig.2 ?

Supporting info:

706-711: suggest merging this short description in the main text, along with figure S8 if possible

Fig S6: add legend for blue and red lines in plots (even if they are described in the caption) as well as labels in Y axis.

We thank the reviewers for their positive comments. Our responses are detailed below in italics.

Reviewer #1 (Remarks to the Author):

I have put most of my comments- both general and detailed into pdf 'post-it notes' in the attached pdf.

In summary, The author list is made up of unparalleled modelling expertise but has less knowledge of the responses - regionally and cumulatively - of marine life to changes (from a physiological and ultimately ecological perspective)

So in some respects it is a premature submission but I think that if carefully revised - and framed more closely to the COP21 scenarios - in particular the 1.5 C warming (and the recent commentary in NCLIM by Magnan et al. it could be a useful starting point for discussions amongst policy makers.

So acceptable after major revisions - that need to get across that although model projections of multiple drivers are well advanced, our understanding of the cumulative effects of multiple drivers lags far behind, as does teasing apart the individual and interactive effects and their mechanistic influence across a wide range of marine life.

Hence a more balanced viewpoint on the range of responses - detrimental, none, beneficial - for the organisms at each trophic level is needed - before we can even contemplate how they will translate ecologically

I sign my reviews - Philip Boyd

The majority of comments from Reviewer 1 can be roughly grouped into 3 categories:

- 1. Climate change-induced responses can be positive or negative*
- 2. The response to combinations of stressors is difficult to predict, with conflicting evidence available*
- 3. The terminology of 'drivers' vs 'stressors' needs to be clarified*

We agree with all of these points. Although our original manuscript attempted to get the first two points across clearly to the reader, we have now amended the text where suggested by the reviewer in the annotated PDF to emphasise these points. We have also carefully gone through the manuscript and checked our use of 'drivers' and 'stressors' to hopefully make our case more clearly.

Where the reviewer's comments in the annotated PDF concern points not covered in the above 3 categories, we respond to them individually below.

skewedness in the title - there will be winners and losers - as is already evident from modelling studies such as Bopp et al. 2013 Biogeosciences -so the manuscript needs to take a more circumspect approach from the start.

We have changed the title to 'Rapid emergence of climate change in environmental drivers of marine ecosystems'.

So in some respects it is a premature submission but I think that if carefully revised - and framed more closely to the COP21 scenarios – in particular the 1.5 C warming (and the recent commentary in NCLIM by Magnan et al. it could be a useful starting point for discussions amongst policy makers.

After COP21, nations submitted Intended Nationally Determined Contributions which indicate their intended level of emissions post-2020. Even if achieved, these intended emissions still imply a median warming of 2.6-3.1 °C by 2100 (Rogelj et al., 2016), within the range of RCP4.5. Therefore our analysis does have relevance to COP21, and we have added comments to make this clear on lines 121-126 and 322-324. We deliberately avoided the use of arbitrary thresholds in our analysis (e.g. 1.5 °C global mean warming), as what is a small change in, e.g. temperature, in one location would be a large change in another. We believe that of more relevance is that conditions exceed the natural variability specific to a particular locale.

As we are still exploring the effects of multi-drivers/stressors on a wide range of organisms - and designing experiments to look at these cumulative effects - see Boyd et al. 2015 NCLIM along with the potential influence of changes in prey (## and quality), acclimation and adaptation we are still a long way from coming up with conclusive regional assessments of the influence of changes in multiple ocean properties - I acknowledge that the authors are cognisant with most of these issues as reflected in their Introduction- but the fledgling state of this research field (and hence the lack of a consensus view) doesnt come across in the ms as written.

As the reviewer notes, we are aware of the difficulty of moving from the kind of analysis presented here to an understanding of ecosystem effects, particularly given the complexity of responses to stressors. The reviewer acknowledges that these issues are reflected in the manuscript's introduction. We have also discussed the difficulties of moving from the kind of analysis presented here to understanding ecosystem response in our discussion. Our aim is not to answer the extremely complex question of how climate trends will affect ecosystems, but to present a quantification of the time of emergence of drivers as a motivation for further exploring the response to climate change. We recognise that a) although 'stress' has negative connotations, the actual organism/ecosystem response to climate change may nevertheless be positive or neutral, b) observations/lab studies of one organism cannot necessarily be scaled-up to an ecosystem response, c) the effect of combinations of stressors is even more poorly understood than the effect of single stressors, d) moving from the kind of analysis presented here to robust predictions of ecosystem response is a 'grand challenge' of our day. We have amended the text of the introduction and discussion (see track changes version of manuscript) to hopefully make our point more clearly.

for example there are many cases in ocean acidification where model projections point to omega values that should dissolve calcifiers yet they subsist in nature - as carbonate chemistry alone cannot capture many of the nuances of organismal physiology - such as proton pumps, organic coatings on calcite - Ries NCLIM 2013 etc - this departure between model projections and observations around multiple ocean properties is likely to be even greater hence this manuscript must be written very carefully

In paragraph 3 of the discussion, we explicitly state that evidence of ocean acidification resulting in long-lasting damage is scarce. Our discussion as a whole brings up the points mentioned above and does not make predictions about what the ecosystem response to climate change will be.

state up front the 4 you consider as this is impt for the differing emergence time of ocean properties
Added.

just one trend?
Changed.

this would be most powerful in a policy frame if it explicitly tackled COP21 scenarios such as 1.5C - its not clear from the abst if it does
See above comment regarding COP21.

the IPCC was a little skewed towards higher trophic levels the energy source /quality for most foodwebs will also rely on nutrients and underwater irradiance - how does the emergence of these factors play out ?
The overall response of the base of the food web to changes in nutrients and light is integrated by primary production. We wished here to represent the overall potential food availability to the marine ecosystem, and hence primary production is the appropriate variable to assess. In addition, primary production has the advantage that it is identified by the IPCC as one of the 4 principal climate drivers in the ocean, so our analysis will therefore readily inform, and be consistent with, IPCC/CMIP analyses in the current and future assessment reports.

simplistic - how will O2 alter oxygen evolving photoautotrophs? their will be regionally nuanced patterns - evident from Bopp et al and many other papers
the idea of positive effects of CC on primary producers is supported by a very small number of studies to date that have looked at all the biologically influential drivers that influence open ocean phytoplankton - such as Boyd et al. NCLIM 2015 - but critically the underlying mechanisms differ from those in the Bopp et al. study /model parameterisation
Our statement that 'the combined effect of these changes is predicted to be an overall global decrease in primary production' is indeed a simplification, but it nevertheless reflects current state-of-the-art model projections. We have nevertheless modified the sentence to read 'Although regionally variable the combined effect of these changes is predicted to be an overall global decrease in primary production'.

Encountered
Changed

again changes in variability versus the mean in the coming decades will influence the relative roles of acclimation versus adaptation – recently detailed in a review that was published after the authors submitted their ms - Boyd et al. 16 GCB but nevertheless the responses of biota with respect to the 'emergence' requires consideration
We have altered this to read 'adapt or acclimate to change' and included the suggested reference.

range has an ambiguous meaning here especially in the context of migration / range shifts etc
We have changed this to read 'push marine ecosystem drivers beyond the range of natural variability'

we arent even close and are still scratching the surface
We agree and have removed 'precise'.

what about heat waves ?? and CC effect on heat waves - recent Wernberg et al. paper in Science

A heat wave is an extreme event and, in keeping with our assumption that trend > natural seasonal range results in stress, this particular event resulted in kelp die-off, i.e. the Wernberg study is entirely consistent with our analysis. We note on line 141-144 that our assessment of extreme events is limited by the temporal resolution of the model output.

but in some cases they can adapt but in others- for longer emergence times they must still acclimate - so its more complex than stated here - see Boyd et al 16 GCB
We have changed this to 'adapt or acclimate'

will they all emerge concurrently?

That is exactly the question the manuscript goes on to pose and then answer as far as possible.

encountering
Changed.

if you are modelling PP then you should also have info on the emergence times for nutrient supply and shifts in underwater irradiance
See above response for why we chose PP.

hotspot for uncertainties ???
Changed.

without analog?
Changed.

needs rebalanced throughout

We have altered the discussion in several places (see track changes version). However, we felt that the discussion was already fairly well-balanced in that nowhere do we claim to know what the ecosystem response may be, or that all stressors will be detrimental. For example, in the 2nd paragraph we discuss whether polar or tropical species may be more vulnerable to future change, but end by saying that on the other hand new niches may become available for species resilient to lower pH and higher temperatures. As another example, in paragraph 3 of the discussion we clearly state that, although the expectation has been that ocean acidification will have a negative effect on calcareous organisms, evidence that OA has a lasting or significant effect is limited. We also acknowledge that translating our results into an understanding of ecosystem response is 'complex in the extreme' (middle paragraph 5) and that our results are merely a 'first step' (end paragraph 6). We also clearly state that our understanding of adaptation capacity (end paragraph 3), migration speed (middle paragraph 4), and the response of ecosystem structure and function (start paragraph 5) is poor. As a result, we conclude (paragraph 6) that 'how individual species will fare, or how the ecosystem as the sum of its parts will fare, is poorly understood' and we recognise that there will be 'winners and losers in the future ocean' (paragraph 6).

driven by a mix of CC and CV - see Edwards et al. 2013 PLOS ONE

We have modified this sentence to 'Smaller motile species, such as zooplankton, have also been observed to migrate in response to climate trends or variability, as in the North Atlantic

where the distribution of warm-water copepod species has shifted northward in recent decades'

Reviewer #2 (Remarks to the Author):

Overall: This is an overall well written, designed, and performed study addressing an important and nontrivial question. It should be of interest to a broader audience and for a high-impact journal. There are a few points where the ms requires some edits and where a few open questions need attention.

Specific Points

Major

1. The language of detection / signal emergence is often unclear. For example, on line 35, it is unclear from what the trend “emerges”? On line 92, the language is “likely to appear”.

We have added on line 36-37 ‘emerge from the background of natural variability’ and changed the sentence on line 99-101 to hopefully make our intent clearer. Other instances have been changed throughout the manuscript.

2. It would help the reader to have an example time series (maybe in the SOM) to illustrate the ToE and other concepts introduced on page 5.

We have added a figure to the supplementary information (Fig. S1).

3. It is unclear / ambiguous (and in my view does not follow from the evidence presented in the ms) that mitigation could “avert” the ecosystem stress (l. 300). Please clarify.

We have altered this to “slow the development of ecosystem stress”.

4. Will the data and analysis code be available for reproduction once the ms is published?

The model output analysed here is publically available via the CMIP5 archive. The data used in the figures (i.e. ToE etc.) will be made available via request.

5. How have the models been fitted (p. 11)?

The models were fitted using generalised least squares, including a first-order autocorrelation term (line 347-348), using R package ‘nlme’. We have added a reference to the relevant package.

6. The ms tests several assumptions and some are violated. How does this impact the probabilistic statements (e.g., l. 342)? Can these statements be tested? How large is the autocorrelation? In general, there is a considerable body of research on detecting trends in field with spatial and/or temporal correlations that the paper is mostly silent on.

The impact on the probabilistic statements is that the chances of a ‘false alarm’ are increased in regions where the assumptions are violated, so that instead of the trend exceeding 95% of the values in the noise, this value would be lower. We have now stated this explicitly on line 375-379. The autocorrelation at lag 1 is of the order 0.5 for all variables. We therefore take into account temporal autocorrelation in our analysis by including an AR(1) error term in the regression (eqn 1), but the reviewer is correct that we do not assess spatial correlation. We are familiar with the literature concerning space-time modelling and exploiting spatial correlation to detect trends (e.g. Bakar and Sahu, 2015, J Stat Soft); this is the focus of our ongoing work.

7. What is the explanation for the bimodal distribution in Figures S6 (a,b)?

The bi-modal distribution indicates that there are two 'families' of climate change pace in SST and PP. This is not unexpected given the heterogeneity of the global ocean and is reflected in the global maps (Figure 1e and h), where, for example, the pace of climate change in SST is very rapid in the Arctic, and rather slower in other regions.

Minor:

1. The citation for IPCC and RCP on lines 110-112 seem to be missing. I assume they refer to Moss et al (2010). In this case, is the 'business-as-usual' a citation? Re-reading the Moss et al (2010) paper suggests to me that interpreting RCP8.5 as business-as-usual can be problematic. Please clarify.

We have included the relevant citation to RCPs and the reference illustrating RCP8.5 as a 'business as usual' scenario.

2. Please define "robustly" and "robust" (l. 284 and 287) or reword. This expression can have very different meanings depending on the discipline of the reader.

We have removed the word 'robust' here.

3. How is a "best course for sustainable management" defined (l. 285)?

We have reworded to 'an appropriate course'.

4. How was the data regridded (l. 313)?

Using linear nearest-neighbour – we've added this information to the methods.

5. Is "extrema" the correct word choice (l. 324)?

This should be 'extreme' – corrected.

6. Please add units to the right colour bar on fig 2.

Figure 2 does not have a right colour bar (the colour bar is located at the bottom of the figure and is labelled).

7. Can the statement on l. 699 indeed be "verified"?

We have changed to 'tested'.

8. Whose / which confidence is discussed on l. 703?

We have altered this sentence.

Reviewer #3 (Remarks to the Author):

Review of "Rapid emergence of climate change in environmental drivers of marine ecosystem stress" by Henson et al.

General comments:

The authors use CMIP5 model output to investigate the emergence of climate change signals and the timing at which annual extremes in key variables (pH, SST, oxygen, primary production) surpasses their natural variability. They find that pH has already been changed beyond it's natural variability, and so has SST in the subtropics and the Arctic. Under a

business-as-usual scenario (RCP8.5), they find that most of the ocean will be subjected to multiple stressors by 2050 (the subtropics and the Arctic by 2030). However, under a mitigation scenario (RCP4.5), the emergence of multiple stressors could be delayed in large parts of the ocean by ~20 years, allowing for longer adaptation periods for ecosystems.

The study is novel and of great interest. It aims at tackling one of the issues related to the vulnerability of marine ecosystems under climate change -the exposure to multiple stressors- and also discusses the relevance of the results in terms of the time that will be available for marine ecosystems to adapt. I found the methods sound and appropriately explained and I believe the conclusions are robust; furthermore, the authors do a good job at presenting some caveats and limitations of their analysis (eg, coastal regions not included in their analysis).

As a potential addition to the manuscript, I would suggest a discussion on how current emissions compare between RCP8.5 and RCP4.5 - given that both scenarios start in 2006, there could be a description of how much closer we are to the business-as-usual scenario (maybe a good place to emphasize the risks of not reducing our emissions ASAP). *Currently, global CO₂ emissions are tracking the RCP8.5 scenario (Peters et al., 2013; Sanford et al., 2014). After COP21, nations submitted Intended Nationally Determined Contributions which indicate their intended level of emissions post-2020. Even if achieved, these intended emissions still imply a median warming of 2.6-3.1 °C by 2100 (Rogelj et al., 2016), within the range of RCP4.5. We have added comments to this effect on lines 121-126 and 322-324.*

Furthermore, I found quite interesting that PP seems to be much more resilient to climate change than other variables that could affect it (eg, SST, pH). Therefore, I would love to see a brief explanation about the expected variability in PP - is it that climate-induced changes in other variables act in different directions, such that PP is not so affected (e.g. warming that would enhance PP overlapped with increased stratification that reduces nutrients and would decrease PP) and/or missing processes in the models (e.g. no phytoplankton groups that actually depend on pH)?.

Primary production seems to be particularly resilient to climate-induced changes, as the ToE is relatively late. PP acts as an integrator of changes in light, temperature and nutrients, so this may reflect compensating effects resulting in a relatively neutral trend. However, Fig S8 demonstrates that the trend in PP is fairly large (~ 5 % per decade, similar to oxygen), suggesting that the noise (i.e. natural variability) is sufficiently large to delay ToE. This is consistent with previous work examining trend vs noise in the context of detecting climate change in observations (Henson et al., 2016), which shows that trends are more rapidly detectable in parameters such as SST and pH than in PP (or nutrients, diatom PP etc.), primarily due to the large natural variability in the latter. The reviewer is also correct that the actual response to change is likely to be far more complex than we can currently predict, as we point out in the manuscript. We have added comments to this effect on line 164-170.

Lastly, the authors mentioned some caveats of the analysis (constrained to use monthly output and miss the potentially larger high-freq variability; the fact that global models do not resolve the coastal ocean, which are regions of higher natural variability). Could the authors discuss any other limitations due to the coarse grids of global models? (eg. missing variability due to eddies and other unresolved mesoscale processes, etc).

We have added a note to that effect on line 144-145.

Overall, I found the manuscript well written and worth of publication in Nature Communications.

Specific comment:

148: "PP and interior oxygen content evince a much slower pace of change". When I look at figure 1, oxygen seems to have a more rapid pace (20-40 years) than PP, except in the tropics. Actually, figure 1h (oxygen) seems as "red" as figure 1e (SST). It does not seem appropriate to equate PP with oxygen (also done in the discussion) - even the ToE for oxygen in the Arctic and OMZ regions seems to be quite close to present date. Also, it may be worth noting that in the OMZ regions, pH+O₂ emerge as stressors by 2030. I recommend adjusting the description of oxygen changes (or consider improving the description and why the authors think is comparable to PP changes).

We have changed this section on lines 165-170.

Minor comments:

51: identify  identified? identifies?

Changed.

116: could use "SST" already

Changed.

183: suggest including reference to fig S8

Added.

188-189: suggest to rewrite: "undavoidable," "unavoidable;" and ", and" ", it"

Changed.

196: I feel "quadruple whammy" is not a friendly phrase for non-native English speakers. I suggest re-phrasing.

This phrase follows from the 'triple whammy' described in Gruber (2011), so we prefer to keep this terminology.

198: "Fig 1" should it read Fig.2 ?

Changed.

Supporting info:

706-711: suggest merging this short description in the main text, along with figure S8 if possible

We agree that Fig S8 should be in the main text, and this now forms Figure 3. The description has been removed from the supplementary material.

Fig S6: add legend for blue and red lines in plots (even if they are described in the caption) as well as labels in Y axis.

We have altered this figure as requested.

Reviewers' comments:

Reviewer #1 (Remarks to the Author):

The authors have generally done a good job of tidying up this manuscript and have also provided clear documentation of their responses to the issues I raised.

there are still a few (mainly minor) amendments required before the ms can be accepted.
line 37 - replace seasonal with climate variability - as seasonal wont capture all of the relevant variability (ENSO etc)

also is there scope within the abstract to indicate whether the 55% is mainly in the low latitude or high latitude ocean as it is rather vague as written.

line 68 - here might be the best place to mention stressors and drivers together to make the point that some effects will be positive - such as shoaling the surface mixed layer in high latitude waters. (c.f. lines 26 and 27 - where each term is mentioned and this could cause some confusion if not revisited).

para commencing line 84

the interplay of drivers can also be antagonistic - for example warming mitigates the effects of acidification on sea urchins - resulting in a zero sum effect. the three examples in this paragraph are all detrimental effects - balance this with a beneficial example?

141 - see prior comment about line 37

170 cant you explore this notion further by teasing part the individual effects of light, temperature and nutrients from the model runs? for example isn't it possible that the slow pace of change could be due to antagonisms between these factors - warming should increase phytoplankton growth rate, as should higher mean underwater irradiances - but these could be offset by reduced nutrient supply. You should be able to dissect these out.

710 some of the text in S-Materials requires attention - such as line 710 etc.

Apologies for the delay in returning this review - the figures and S-materials/S-figures represented a considerable amount of reading

Philip Boyd

Reviewer #2 (Remarks to the Author):

General comments:

This is a mostly responsive revision that addressed many of the comments. However, there are several instances where the responses did not address the raised question and/or issues. These points need to be carefully addressed before the ms raises to a level of quality one expects for this outlet. These points can be easily addressed, if the authors choose to do so. I would be happy to re-review a suitably revised manuscript.

Specific points:

1) The review asked: "will the data and analysis code be available for reproduction once the ms is

published"? You respond that the derived data for the figures will be made available on request. This does not address the question regarding the code. Please address the questions raised in the review and add the relevant information to the ms in the revision process. Your response could be interpreted to imply that the code may not be distributed. This would then be in contradiction to my understanding of the Nature policy (see: <http://www.nature.com/authors/policies/availability.html>).

2) The review asked to please explain / check whether an interpretation of the RCP8.5 scenario as a "business-as-usual" is appropriate. The revision changed the wording a bit and added two references. Alas, on inspection, I could not find evidence for this interpretation in the two cited references. Please either revise the wording carefully or provide a clear citation from the original literature.

Reviewer #3 (Remarks to the Author):

I appreciate the authors' efforts to include my comments and those of other two reviewers and think that the result is an even stronger manuscript.

The only comment I have left is that I do not always agree with the current use of "driver" instead of "stressor"; I found that in many cases the new use of "driver" is not appropriate. For instance, line 118-119 of the annotated manuscript now reads "at which multiple drivers emerge", but drivers are always drivers, they don't really "emerge" - their extreme/untolerable/out-of-natural-range changes do "emerge". This new use of "driver" (aimed at replacing the word "stressor") is found in many other places in the manuscript, e.g. lines 175, 177, 199, 214, etc of the annotated ms (this list is not complete). I haven't been able to access the commented pdf of reviewer 1 to fully understand his concern about the "drivers" vs "stressors" terminology, but I would recommend to clearly define what the authors mean by "stressor" (eg, when the annual extreme exceeds the natural variability) and keep the use of this word where appropriate. In some cases (eg line 211 or line 218), I think that the new use of driver is appropriate. I recommend the paper for publication after a thorough revision of the use of "driver" vs "stressor".

We thank the reviewers for their confirmation that we have addressed their previous comments. Our responses to the additional comments are in italics below. In the tracked changes document the original changes are marked in red, and the new changes in response to the 2nd round of reviews are in blue. Line numbers below refer to the PDF of the manuscript without track changes.

Reviewer #1 (Remarks to the Author):

The authors have generally done a good job of tidying up this manuscript and have also provided clear documentation of their responses to the issues I raised.

there are still a few (mainly minor) amendments required before the ms can be accepted.
line 37 - replace seasonal with climate variability - as seasonal wont capture all of the relevant variability (ENSO etc)

also is there scope within the abstract to indicate whether the 55% is mainly in the low latitude or high latitude ocean as it is rather vague as written.

We have changed lines 35-39 to read: “By analysing an ensemble of models we find that, within the next 15 years, the climate change-driven trends in multiple ecosystem drivers emerge from the background of natural variability in 55% of the ocean (principally at low latitudes and in the Arctic) and propagate rapidly to encompass 86% of the ocean by 2050 under a ‘business-as-usual’ scenario.”

line 68 - here might be the best place to mention stressors and drivers together to make the point that some effects will be positive - such as shoaling the surface mixed layer in high latitude waters. (c.f. lines 26 and 27 - where each term is mentioned and this could cause some confusion if not revisited).

We have added lines 71-73 to clarify this point: “However, in some cases positive (or neutral) responses to potential marine stressors have been observed (12, 13), implying that uncertainty surrounding the future of the marine ecosystem is large. Here we adopt the terminology that a change in a potential stressor where the ecosystem response is unknown (and may not necessarily be negative) is termed a ‘driver’.”

para commencing line 84

the interplay of drivers can also be antagonistic - for example warming mitigates the effects of acidification on sea urchins - resulting in a zero sum effect. the three examples in this paragraph are all detrimental effects - balance this with a beneficial example?

We have included the example suggested by the reviewer so that lines 98-102 now read, “Other studies have shown, however, that multiple stressors can interact in unexpected ways, sometimes resulting in a positive or neutral effect (12, 28). For example, warming and ocean acidification have antagonistic effects on sea urchin larval growth, resulting in minimal overall impact (although both have a negative effect on larval abnormality(29)).”

141 - see prior comment about line 37

We clearly define here our definition of natural variability for the purposes of this manuscript (line 144-145): “we define natural variability using the seasonal amplitude derived from monthly model output.”

170 cant you explore this notion further by teasing part the individual effects of light, temperature and nutrients from the model runs? for example isn't it possible that the slow pace of change could be due to antagonisms between these factors - warming should increase phytoplankton growth rate, as should higher mean underwater irradiances - but these could be

offset by reduced nutrient supply. You should be able to dissect these out.

The separate effects of light, temperature and nutrients on PP is an interesting topic, but one we feel would be a distraction from the main message of this manuscript. PP is the ultimate food source for life in the ocean and thus is the relevant parameter to use in an assessment of the emergence of long-term trends in factors likely to affect marine ecosystems. Precisely which element drives the trend in PP is less relevant here. However, what the reviewer suggests would make an interesting future study. Some work in this direction has already been done globally (Laufkötter et al. 2015, Biogeosciences) and for the Arctic specifically (Popova et al. 2012, JGR).

710 some of the text in S-Materials requires attention - such as line 710 etc.

We believe the reviewer is referring to the use of 'driver' vs 'stressor'. We have changed line 722 and figure captions.

Reviewer #2 (Remarks to the Author):

General comments:

This is a mostly responsive revision that addressed many of the comments. However, there are several instances where the responses did not address the raised question and/or issues. These points need to be carefully addressed before the ms raises to a level of quality one expects for this outlet. These points can be easily addressed, if the authors choose to do so. I would be happy to re-review a suitably revised manuscript.

Specific points:

1) The review asked: "will the data and analysis code be available for reproduction once the ms is published"? You respond that the derived data for the figures will be made available on request. This does not address the question regarding the code. Please address the questions raised in the review and add the relevant information to the ms in the revision process. Your response could be interpreted to imply that the code may not be distributed. This would then be in contradiction to my understanding of the Nature policy (see: <http://www.nature.com/authors/policies/availability.html>).

This was an oversight on our part – apologies. The code will also be made available on request.

2) The review asked to please explain / check whether an interpretation of the RCP8.5 scenario as a "business-as-usual" is appropriate. The revision changed the wording a bit and added two references. Alas, on inspection, I could not find evidence for this interpretation in the two cited references. Please either revise the wording carefully or provide a clear citation from the original literature.

The cited paper by Riahi et al. (2011) discusses the socio-economic and emissions characteristics of the RCP8.5 and states that the RCP8.5 is a 'business-as-usual' scenario. The first sentence of the discussion states, "RCP8.5 depicts, compared to the scenario literature, a high-emission business as usual scenario." We have also added a reference to Magnan et al. (2016), published in the Nature group of journals, which repeatedly refers to the RCP8.5 scenario as 'business-as-usual'.

Reviewer #3 (Remarks to the Author):

I appreciate the authors' efforts to include my comments and those of other two reviewers and think that the result is an even stronger manuscript.

The only comment I have left is that I do not always agree with the current use of "driver" instead of "stressor"; I found that in many cases the new use of "driver" is not appropriate. For instance, line 118-119 of the annotated manuscript now reads "at which multiple drivers emerge", but drivers are always drivers, they don't really "emerge" - their extreme/untolerable/out-of-natural-range changes do "emerge". This new use of "driver" (aimed at replacing the word "stressor") is found in many other places in the manuscript, e.g. lines 175, 177, 199, 214, etc of the annotated ms (this list is not complete). I haven't been able to access the commented pdf of reviewer 1 to fully understand his concern about the "drivers" vs "stressors" terminology, but I would recommend to clearly define what the authors mean by "stressor" (eg, when the annual extreme exceeds the natural variability) and keep the use of this word where appropriate. In some cases (eg line 211 or line 218), I think that the new use of driver is appropriate. I recommend the paper for publication after a thorough revision of the use of "driver" vs "stressor".

The crux of reviewer 1's preference for 'driver' over 'stressor' in some sentences was based on the argument that 'stressor' implies a negative effect, whereas 'driver' doesn't carry negative connotations and so should be used where the effect of a change on the ecosystem isn't known. Generally we agree with this argument and so have changed the phrasing of the manuscript accordingly. We've clarified our terminology on lines 71-73 which now read, "Here we adopt the terminology that a change in a potential stressor where the ecosystem response is unknown (and may not necessarily be negative) is termed a 'driver'."